# Rheological and Self-Healing Behavior of Hydrogels Synthesized from l-Lysine-Functionalized Alginate Dialdehyde

**DOI:** 10.3390/polym15041010

**Published:** 2023-02-17

**Authors:** Arlina Prima Putri, Ranjita K. Bose, Mochamad Chalid, Francesco Picchioni

**Affiliations:** 1Department of Chemical Engineering—Product Technology, University of Groningen, Nijenborgh 4, 9747 AG Groningen, The Netherlands; 2Metallurgical and Material Engineering Department, Universitas Indonesia, Depok 16424, Indonesia

**Keywords:** alginate, alginate dialdehyde, hydrogel, shear-thinning, self-healing

## Abstract

Alginate dialdehyde and l-lysine-functionalized alginate dialdehyde were prepared to provide active aldehyde and l-lysine sites along the alginate backbone, respectively. Different concentrations of substrates and the reduction agent were added, and their influence on the degree of l-lysine substitution was evaluated. An amination reduction reaction (with l-lysine) was conducted on alginate dialdehyde with a 31% degree of oxidation. The NMR confirmed the presence of l-lysine functionality with the degree of substitution of 20%. The structural change of the polymer was observed via FTIR spectroscopy, confirming the formation of Schiff base covalent linkage after the crosslinking. The additional l-lysine sites on functionalized alginate dialdehyde provide more crosslinking sites on the hydrogel, which leads to a higher modulus storage rate than in the original alginate dialdehyde. This results in dynamic covalent bonds, which are attributed to the alginate derivative–gelatin hydrogels with shear-thinning and self-healing properties. The results suggested that the concentration and stoichiometric ratio of alginate dialdehyde, l-lysine-functionalized alginate dialdehyde, and gelatin play a fundamental role in the hydrogel’s mechanical properties.

## 1. Introduction

Alginate has played an essential role in pharmaceuticals as an excipient and biomolecule matrix in drug products [1]. The applications have ranged from creating a controlled system delivery, specific targeted delivery, or delivering special packages, such as a bacteriophage [2,3,4]. Alginate consists of mannuronic (M) and guluronic (G) acid residues arranged in an irregular blockwise pattern along the linear chain [5]. However, a lack of alginate-degrading enzymes leads to challenges in removing high molecular weight residues from the body through excretion by the kidneys [6,7]; hence, modified alginate has become more preferred. One alginate derivate that has drawn much research attention is alginate dialdehyde (ADA). ADA is a partially oxidized alginate that presents more reactive groups, greater cell interactions, less structure rigidity, and fast in vivo degradability [7].

ADA is often associated with the gelation process that occurs in the presence of crosslinking agents. For example, hydrogels of ADA can be achieved via crosslinking with gelatin in the presence of borax [8]. In this method, the concentration of aldehyde groups along the backbone is essential. An available aldehyde in ADA facilitates crosslinking with gelatin through Schiff’s base formation with the ε-amino groups of lysine or hydroxylysine residues of gelatin [9]. This ADA–gelatin hydrogel has been reported to be used in various applications, such as for encapsulation and tissue engineering and in bio-printed scaffolds [10,11,12]. The biodegradability of this hydrogel is tunable by controlling the degree of ADA oxidation or by arranging a specific ratio between the ADA and gelatin [8,13]. However, previously this Schiff’s base crosslinked hydrogel method was mainly utilized for unmodified ADA [10,11,14]. Additionaly, this hydrogel formation method is rapid, straighforward, and does not require supllementary toxic chemical crosslinking agents [15].

Nowadays, increasing interest has been focused on the functionalization of ADA. This modification provides active sites for cell attachment and bone tissue scaffolds, meaning it can serve as a surfactant and sustained release matrix for drug delivery (Table 1). Functionalized ADA is also used in hydrogel form to serve as a sustained release matrix. Since oxidation mainly affects the G blocks of alginate, it reduces the cooperative ability for divalent ion interactions [16]. For example, Β-cyclodextrin-grafted ADA (8% oxidation) was able to form a hydrogel with calcium chloride, but it was very labile and easy to deform [17]. In this case, a hydrogel synthesized with a crosslinking agent is more suitable for functionalized ADA, since the additional crosslinker binds individual chains with one another to form a network [18]. Thus, studying the gelatination of functionalized ADA through crosslinking with the gelatin is necessary. Moreover, understanding the mechanical properties of this hydrogel is also required.

Shear-thinning and self-healing hydrogels have been investigated for various biomedical applications. Such hydrogels allow homogenous encapsulation, clogging-free injection through needles, and an ability to recover to their initial state [30]. A dynamic covalent chemistry interaction is needed to form the shear-thinning and self-healing hydrogels. This interaction can be provided by the Schiff base reaction, which occurs in the crosslinking formation between ADA and gelatin. Such interactions are formed by a pseudocovalent bond, a highly stable yet physiologically degradable gel network that improves the cell adhesion, biocompatibility, and biodegradability [31]. The self-healing properties of hydrogels are due to their ability to reform hydrogen bonds and for electrostatic repulsion reversibly [32].

Here, we report on the synthesis of an l-lysine-functionalized ADA (f-ADA) hydrogel with gelatin as the crosslinking agent. The side chain of l-lysine in the f-ADA will served as the additional linker to allow more crosslinking with gelatin than the original ADA. The functionalization of ADA was achieved via reductive amination, the protocol for which was established using l-lysine as the substituent and NaBH_4_/SiO_2_ as the reduction agent. In the gelatinization process, a gelatin solution was utilized to form dynamic crosslinking hydrogels. In this work, we performed a series of rheological tests to determine the viscosity and storage and loss moduli and to assess the shear-thinning and self-healing properties of the hydrogels [33]. The rheological studies showed the shear-thinning and self-healing behaviors, which support the application of these hydrogels for biomedical applications such as drug delivery. We considered the different mechanical responses along with the variations in the ADA, l-lysine, and gelatin contents in hydrogels. The further modification of ADA with l-lysine resulted in hydrogels with high storage moduli. All hydrogels showed an ability to self-heal via a strain recovery test, and the four selected hydrogel monoliths were mended after seven days of resting. Our finding will be useful for the application of functionalized alginate dialdehyde–gelatin hydrogel system as a candidate for a drug delivery platform.

## 2. Materials and Methods

### 2.1. Materials

The sodium alginate (a sodium salt of alginic acid from brown algae), sodium (meta)periodate, sodium borohydride, silicon dioxide (325 mesh), l-lysine, gelatin from bovine skin (type B), phosphate-buffered saline, sodium chloride, sodium hydroxide, and ethylene glycol were purchased from Sigma-Aldrich Chemie N.V., Zwijndrecht, the Netherlands. The ethanol, *n*-propanol, and acetone were purchased from Avantor Performance Materials B.V., Dordrecht, the Netherlands.

### 2.2. Methods

#### 2.2.1. Synthesis of the Alginate Dialdehyde

The ADA was synthesized as previously described [16]. Briefly, 2 g of sodium alginate and 0.536 g of sodium metaperiodate were dispersed in 25 and 20 mL of MQ water, respectively. The sodium metaperiodate solution was then slowly added to the sodium alginate solution, and *n*-propanol 10% (*v*/*v*) was added as a radical scavenger. The reaction was performed at room temperature under dark conditions with continuous stirring for 48 h. It was then quenched with 1 mL of ethylene glycol and left under stirring for another 30 min. The ADA was precipitated by adding NaCl (1.00 g) with ethanol (300 mL). This process was repeated three times for reprecipitation. The residue was then dried in a vacuum oven at 40 °C until reaching a constant weight.

#### 2.2.2. Determination of the Oxidation Degree

The calculation of the degree of oxidation was achieved via hydroxylamine hydrochloride titration according to the preceding experiment [34]. Here, 25 mL of hydroxylamine hydrochloride 0.25 N solution containing methyl orange solution was prepared, then a 0.1 g sample was added. After stirring for 2 h, the mixture was titrated with sodium hydroxide at 0.1 N until the red-to-yellow was endpoint reached, and the pH was 4. The aldehyde content was determined by Equation (1) [13]:(1)$$DO_{x}=\frac{(V_{NaOH_{sample}}-V_{NaOH_{blank}})\times N_{NaOH}}{m_{sample}\times MW_{alginate\;repeating\;unit}}$$
where *V* is the consumed volume of sodium hydroxide by the ADA (sample) and sodium alginate (blank) in L, the normality of the sodium hydroxide is 0.1 N, m is the mass of the sample (g), and the molecular weight of the alginate repeating unit is 198 g/mol. The experiment was conducted in triplicate, and the reported values are the averages.

#### 2.2.3. Reductive Amination of ADA with l-lysine

The ADA and l-lysine were prepared in MQ water corresponding to a final concentration of ADA 0.5 mmol, and the l-lysine contents varied from 0.25 mmol to 1 mmol. The ADA solution was poured into a vial with 2, 6, or 10% (*w*/*v*) SiO_2_. The mixture was stirred, and an equal volume of the amino acid was gradually added to the final volume of 5 mL, with 10% (*v*/*v*) of MeOH. The NaBH_4_ powders with a molar ratio 1:3:5 to oxidized alginate were added slowly to the mixture. The pH was adjusted to the desired pH of 5.8 using 1 M acetate buffer. The reaction was left overnight at room temperature under stirring. At the end of the reaction, the SiO_2_ was separated via vacuum filtration. The solutions were then subjected to dialysis (MWCO: 3500 Da), followed by lyophilization.

The optimized protocol was used to scale-up the production of f-ADA for further studies, providing sufficient hydrogel and subsequent characterization materials.

#### 2.2.4. NMR Spectroscopy

The alginate samples for NMR spectroscopy were prepared via mild acid hydrolysis as described in [35]. Each 10 mg of hydrolysis sample was dissolved in 600 µL D_2_O (99.9%) containing 0.05%wt of the internal standard 3-(trimethylsilyl) propionic 2,2,3,3-d4 acid and sodium salt (Sigma-Aldrich, Zwijndrecht, the Netherlands). The NMR experiments were performed on a Variant Inova 500 MHz instrument with a pulse-field gradient Performa I (20 G/cm) reaction flow system and a Bruker Avance NEO 600 MHz (Billerica, MA, USA) instrument with a pulse-field gradient amplifier and GRASP II10A spectrometers, each equipped with a 5 mm PFG SW probe and 5 mm CyroProbe prodigy BBO, respectively. The degree of substitution was determined by the 1D 1H spectra from NMR at 500 MHz, recorded at 90 °C and with a 25 s delay between measurements (90° angle and 256 scans). The diffusion-ordered spectroscopy (DOSY) spectra were measured using a stimulated echo pulse sequence with bipolar gradients (STEBPGP) recorded with a 500 ms diffusion delay assessing the diffusion coupling of products (600 MHz, 25 °C). The recorded spectra were later processed and analyzed with MNova 12.0.1 software. The DS was estimated from the integration ratio of benzylamine aromatic protons (δ 7.55 ppm) compared to the anomeric protons in the alginate (region δ 4.3–5.3 ppm) following Equation (2) [36]:(2)$$DS\;\%=\frac{I_{subs}/5}{I_{alg}/3}$$
where *I_subs_* is the integrated peak area of the l-lysine signal and *I_alg_* is the integrated peak area of the alginate residue. The diffusion-ordered spectroscopy (DOSY) spectra were measured using a stimulated echo pulse sequence with bipolar gradients (STEBPGP) recorded with a 500 ms diffusion delay assessing the diffusion coupling of products (600 MHz, 25 °C). The spectra were processed and analyzed with MNova 12.0.1 software.

#### 2.2.5. Gel permeation Chromatography (GPC)

The molecular weight of the alginate samples was obtained via aqueous gel permeation chromatography using an Agilent 1200 system equipped with a differential refractive index (DRI) detector and polymer standard service (PSS) column set (PSS SUPREMA 100 Å, 1000 Å, 3000 Å). A solution of 0.05 M NaNO_3_ was used as the mobile phase. The system temperature was set to 40 °C, and a flow rate of 1 mL/min was used. The spectra were calibrated using pullulan with 1% ethylene glycol as an internal standard. The alginate samples were diluted with ethylene glycol to obtain a 1% solution. Before the analysis, all samples were filtered through a 0.2 µm pore membrane. All samples were analyzed twice, and the average values are reported in this study.

#### 2.2.6. ATR-FTIR Spectroscopy

The infrared spectra of all alginate samples were recorded using Fourier transform infrared (FT-IR) spectroscopy with an attenuated total reflection (ATR) sampler on an IRTracer-100 Shimadzu instrument (Kyoto, Japan). All data were processed with LabSolutions IR software. The measurements were conducted in the range of 600–4000 cm^−1^ at room temperature using 64 scans and a resolution of 4 cm^−1^. The reported data are the results from the represented samples. 

#### 2.2.7. Elemental Analysis

An elemental analysis was performed to determine the amounts of nitrogen in the alginate samples using an automated Euro Vector EA3000 CHNS analyzer with acetanilide as a calibration reference. All samples were analyzed twice, and the average values are reported in this study.

#### 2.2.8. Schiff Base Hydrogel Preparation

The alginate–gelatin hydrogels were prepared as previously described [13]. 15, and 20% solution of alginate samples were prepared in 10× PBS. The alginate solutions were then mixed with an equal volume of 10, 15, and 20% solution of gelatin to the final volume of 5 mL in MQ water, and their compositions are shown in Table 2. The mixture was reacted in a scintillation vial, stirring in a water bath at 37 °C for 120 min. The hydrogels were kept at 6 °C prior to analysis.

#### 2.2.9. Determination of the Hydrogel’s Crosslinking Degree 

The degree of crosslinking of alginate–gelatin hydrogels was determined by NHN assay [37]. The alginate–gelatin hydrogel samples (1 ± 0.1 g) were heated to 100 °C in an oil bath with 1 mL of 2% (*w*/*v*) ninhydrin solution for 20 min. The solution was then cooled down to room temperature before dilution with 5 mL of isopropanol 50% (*v*/*v*). The optical absorbance was recorded at 570 nm using a UV-Vis spectrophotometer (Aquamate, ThemoSpectronic, Waltham, MA, USA). The number of free amino groups in the crosslinked hydrogels after the ninhydrin reaction was proportional to the optical absorbance of the solution. A standard curve of the glycine concentration vs. absorbance was used to determine the free NH_2_ group concentration in the crosslinked hydrogels. The ADA–gelatin blend samples without crosslinking were used as a control. The degree of crosslinking was calculated following Equation (3) [37]:(3)$$\mathrm{Degree}\;\mathrm{of}\;\mathrm{crosslinking}\;\left(\%\right)=\frac{{\left({\mathrm{NH}}_{2}\right)}_{\mathrm{nc}}-{\left({\mathrm{NH}}_{2}\right)}_{c}}{{\left({\mathrm{NH}}_{2}\right)}_{\mathrm{nc}}}\times 100$$
where (NH_2_)_nc_ is the mole fraction of free amino groups in the non-crosslinking sample and (NH_2_)_c_ belongs to the crosslinking sample. The experiment was conducted in triplicate, and the reported values are the averages.

#### 2.2.10. Rheological Studies

The rheological measurement of hydrogels was carried out with a HAAKE MARS III (ThermoScientific, Bleiswijk, the Netherlands) rheometer equipped with a 20-mm-diameter cone (1° angle) and plate geometry. The hydrogel sample was molded in a syringe with a 1 mm diameter. The hydrogel was pushed out easily from the syringe after it was set and it was cut with sterile surgical blades with a ± 1 mm thickness. Then, it was carefully placed on the lower plate. After the upper plate was brought into the measurement position, the excess hydrogel was removed from the plate edges. The samples were incubated in the measuring system for 5 min before measurement. All measurements were carried out at 20 °C unless otherwise mentioned, and a 0.052 mm gap was used.

The determination of the viscosity curves was carried out in controlled shear rate mode by logarithmically increasing the shear rate from 0.05 to 500 s^−1^. The linear viscoelastic region (LVE) was obtained via amplitude strain sweeps with the deformation ranging from 10^−1^ to 10^−2^ at a frequency of 1 Hz. All rheological experiments were conducted in triplicate and the reported data are the average values. 

#### 2.2.11. Self-Healing Studies

The thixotropic properties were studied using the rheological method mentioned above. Each loop test consisted of 3 steps: (1) the application of 1% strain at a frequency of 1 Hz for 5 min; (2) an increase in the shear strain to 400% to ensure the rupture of the gel, which was maintained for 2 min; (3) a reduction in the same rate to the initial strain, which was kept for 15 min to stabilize the gel network recovery. During these steps, the gel behavior was characterized by its storage and loss moduli (*G′* and *G″*). 

Hydrogel monoliths were used to study the effect of l-lysine functionalized on ADA on the self-healing capacity and duration of healing. Monolithic hydrogels were obtained and molded in a 2 mL syringe with the tip cut off. The prepared monolithic hydrogels were cut into three parts and put back together with the fracture surfaces facing each other side-by-side. The monoliths were placed in a closed Petri dish, stored at 6 °C, and left to rest for seven days.

## 3. Results and Discussion

Gel permeation chromatography (Table 2) and ^1^H NMR spectroscopy (Appendix A) were performed to characterize the molecular weight and chemical structure of sodium alginate before oxidation. The composition of the uronate residues in sodium alginate was found to characterize the substrate. The anomeric region signals of δ 5.07, 4.70, and 4.46 ppm were assigned to H_1_-G, H_1_-M and H_5_-GM, and H_5_-GG, respectively [5]. Within the set limit for each anomeric signal, the M/G ratio at a value of 1.29 was obtained [36]. These data were confirmed by a previous study, which reported that the M/G ratio for purified commercial alginate was 1.3 [38,39,40].

### 3.1. Synthesis of ADA

The partial oxidization of alginate was successfully achieved using sodium metaperiodate. Hydroxyl groups of carbon 2 and carbon 3 of the repetitive unit of alginate were oxidized, leading to the formation of two aldehyde groups (Appendix A). The ATR FTIR data displayed the new aldehydes groups that formed (Figure 1). The results of the GPC and potentiometric titration of ADA are shown in Table 2. The decrease in molecular weight indicates that the oxidation resulted in the depolymerization of the alginate. This finding is in agreement with the previous study, where the depolymerization of alginate occurred after the oxidation reaction [41].

### 3.2. The Influence of the Substrates and Reduction Agents on the Degree of Substitution of ADA

The reduction of alginate dialdehyde was carried out at pH 5.8 at room temperature, utilizing ADA and l-lysine as the substrates and NaBH_4_/SiO_2_ as the reduction agent. For a better understanding of the influence of each compound, nine cycles of the reaction were conducted. The degree of substitution was determined to specify the efficiency of each reaction. ^1^H NMR was chosen to calculate the substitution degree of l-lysine to ADA, and the data are shown in Figure 1 (Appendix A).

Despite the introduction of aldehydes at C_2_ and C_3_, only a fraction of those groups was available for reaction with l-lysine due to the hemiacetal formation of free aldehydes groups [7]. Consequently, the 2:1 molar ratio of ADA to l-lysine provided a higher substitution degree than an excess of the substituent (Figure 2A). A higher amount of NaBH_4_ provides a higher hydride concentration in aqueous reactions under acidic conditions [42]. Figure 2B shows that the ADA reductive amination reaction’s substitution degree increased with the increase in the sodium borohydride molar ratio. To compare the formation of covalent bonds between the different ratios of ADA and NaBH_4_, NMR DOSY was conducted. The DOSY results show that the formula of 2:1:10 ADA/l-lysine/NaBH_4_ provided the same region of diffusion of l-lysine and ADA by means of the covalent linkage that was formed (Appendix A). The presence of SiO_2_ in reducing the carbonyl compound with NaBH_4_ was found to accelerate the rate of reduction [43]. However, it is unclear how the different amounts of SiO_2_ will affect the result. The ^1^H NMR data (Figure 2) showed that the substitution degree range was between 4 and 22%. In conclusion, increasing the ADA/NaBH_4_ and lowering the ADA/l-lysine molar ratio raised the substitution degree of l-lysine to ADA (Figure 2A,B). Despite using a 2:1:10 ADA/l-lysine/NaBH_4_ molar ratio, which provided the highest substitution degree, the 2:1:6 formula with 6% (*w*/*v*) SiO_2_ was selected for further studies. Since from Figure 2C we can concluded that 6% of SiO_2_ resulted the highest degree of substitution.

### 3.3. Synthesis of l-Lysine-Functionalized ADA

Based on the influence of the substrates and the reduction agents on the degree of substitution studies, the coupling of l-lysine to ADA at a molar ratio of 2:1 (ADA/l-lysine) was conducted to synthesize the l-lysine−functionalized ADA. This reductive amination reaction was utilized at an ADA−to−NaBH_4_ molar ratio of 2:6 and with 6% (*w*/*v*) of SiO_2_. In Figure 3A, the ^1^H NMR spectrum of the l-lysine−functionalized ADA is shown. Based on this spectrum, the degree of substitution at 20.1% was calculated with a nitrogen percentage of 2.47% (Appendix A). As seen in Table 2, both the periodate oxidation and reductive amination reaction lead to alginate depolymerization. The first reaction results in glycol cleavage on the alginate structure, thereby promoting a more flexible structure of the ADA, which leads to depolymerization after the second reaction [44].

NMR DOSY was employed to verify the covalent bonding between ADA and l-lysine (Figure 3B). The diffusion coefficient of the covalent bond of l-lysine−functionalized ADA appeared in the same region. For comparison, the physical mixture of ADA and l-lysine, which was stirred for 24 h without the reduction agent, was diluted in D_2_O before measuring the DOSY spectrum (Figure 3C). This mixture signal showed the difference between the ADA and l-lysine diffusion coefficients. The DOSY spectrum difference between the covalent bonding and physical mixture of ADA and l-lysine indicated that the coupling was successful.

### 3.4. Hydrogel of l-Lysine-Functionalized ADA with Gelatin

Hydrogels of alginate dialdehyde–gelatin have been intensively studied for diverse biomedical applications. This hydrogel was reported to be suitable for tissue engineering and drug delivery injection systems [8]. On the other hand, alginate dialdehyde also reacted further to the coupling with bioactive molecules [20]. Therefore, the crosslinking efficiency of reductive alginate dialdehyde coupling with l-lysine, a model compound for bioactive substances, was further investigated here.

The degree of crosslinking of all alginate hydrogels is reported in Table 3. A higher degree of crosslinking was observed in hydrogels with 20% (*w*/*v*) gelatin. The greater the gelatin concentration in the solutions, the greater the interaction between the ε-amino acid groups and the free aldehydes in the ADA and f−ADA. The crosslinking interaction was correlated to the moles of reacted ε-amino acid groups of lysine or hydroxylysine [14]. It was reported that in the fabrication of alginate–gelatin hydrogels, the aldehyde end groups of the alginate chain end contributed more to the crosslinking reactions than the aldehyde groups along the chain [13]. In this study, the same amino acid, l-lysine, was used to modify ADA to establish a much lower molecular weight of ADA with fewer aldehyde groups along the chain but more aldehyde end groups, because lower molecular weight alginate derivatives lead to a higher crosslinking probability, as already reported before [13]. 

The ATP−FTIR spectra of sodium alginate, ADA, f−ADA, gelatin, ADA15GEL15, and LYS15GEL15 are shown in Figure 1. The half−full spectra of all alginate samples are presented in Figure 1 and the full spectra are presented in Appendix A. The spectra show the characteristic absorption bands of the polysaccharide structure in all alginate samples. The FTIR spectra of all alginate samples displayed a broadband at 3259 cm^−1^, indicating the presence of hydroxyl groups (Appendix A). The carboxylate asymmetric and symmetric stretching vibrations were observed at 1595 and 1406 cm^−1^, and C−O−C stretching at 1026 cm^−1^ [45]. The bands at 1082 and 818 cm^−1^, which were assigned to symmetrical C−O−C stretching, were reduced due to chain cleavage [13]. A new band at 1721 cm^−1^ was detected in the ADA spectrum, confirming the aldehydes groups [45]. This new aldehyde group absorption peak then disappeared, and a new peak appeared at 1256 cm^−1^, attributed to C-N stretching of f−ADA as a result of reductive amination [46].

The spectra of the hydrogels are shown in Figure 3B. The ADA15GEL15 and LYS15GEL15 were observed with alginate characteristic peaks (dash). The absorption bands at 1630 and 1526 cm^−1^ were assigned to the gelatin’s N−H stretching vibration from the amide [47]. The spectra of ADA15GEL15 and LYS15GEL15 presented absorption bands at 1624, 1545, 1614, and 1556 cm^−1^, which were assigned to ν(C=N) and suggested Schiff base formation [14]. The absence of a 1526 cm^−1^ peak for both hydrogel samples was assigned to the amide group. These findings suggested that this amide group was involved in the crosslinking reaction [14].

To analyze the differences in the morphology of these alginate hydrogels, scanning electron microscopy (SEM) was collected from ADA and l-lysine-functionalized ADA (f-ADA) hydrogels. The image showed that the hydrogel after the functionalization with l-lysine had bigger pores than the ADA hydrogels (Appendix A).

### 3.5. Shear-Thinning Characteristic

Shear-thinning and self-healing hydrogels have been investigated for biomedical applications such as drug delivery. Such properties can be observed rheologically, where shear-thinning hydrogels exhibit a viscous flow under shear stress and recover when the applied stress is removed [30]. 

The flow curve describes the rheological behavior of the hydrogel, particularly its viscosity dependency over the applied shear rate. In Figure 4, the shear-thinning behavior of all hydrogel samples is shown. This behavior is an essential characteristic of a hydrogel that can be valuable in its application, such as for injection and drug delivery [48]. In general, it can be seen that the viscosity of the hydrogel samples was affected mainly by the concentration of gelatin. All four hydrogels containing a 20% gelatin solution showed the highest viscosities at close to a zero shear rate, indicating that higher amounts of gelatin in the hydrogels increased the degree of crosslinking, which was reflected in the increase in viscosity. 

The frequency sweep tests were performed within the LVE, which was determined before this step, with increasing frequency rates ranging from 10^−2^ to 10^−1^ Hz (Appendix A). The storage (*G′*) and loss (*G″*) moduli were determined to monitor the mechanical properties of the viscoelastic materials. *G′* is measured from the energy stored during the loading cycle, indicating its solid or elastic character, while *G″* indicates the amount of energy dissipated during the loading cycle. A combination of both parameters gives insights into the Newtonian behavior of the hydrogels, viscoelastic behavior, and sol–gel transformation point [49]. Thus, according to Figure 5 and Appendix A, *G′* in all samples is greater than *G″*, indicating that all samples have solid-like behavior. In all hydrogel samples, the value of these moduli was likely determined by the amount of gelatin in the hydrogel formula. Thus, two samples, ADA15GEL20 and LYS15GEL20, had significantly higher *G′* values compared to the other samples, indicating that both samples have the most solid-like behavior. Comparing Figure 5A to Figure 5B, it is noticeable that the LYS GEL samples have higher storage modulus points (400–1150 Pa) than the ADAGEL samples (100–1000 Pa). This shows that the molecular weight of the alginate derivatives and l-lysine residue influence the modulus measurement. A lower molecular size reduces the entanglement between the polymer chains and provides more areas for crosslinking between alginate fragments and gelatin, leading to more crosslinking in the hydrogel sample.

### 3.6. Self-Healing Properties

Self-healing hydrogels are not vulnerable to stress-induced crack formation, which could prolonging their lifetime. This is due to the intrinsic dynamic equilibrium between the aldehyde and the amine reactants, meaning the interactions can be considered pseudocovalent [30,31]. In order to differentiate the formation of the covalent bond between aldehyde groups and l-lysine residues of ADA and f−ADA, the thixotropic behavior of crosslinked hydrogels was also investigated using a rheological loop test. The results indicated that a hydrogel’s self-healing behavior is mainly due to the amount of gelatin used as the crosslinker agent. The hydrogel recovery rates after the 3−step loop test ranged from 40 to 95% (Figure 6).

In order to study the effect of crosslinking on the self-healing of Schiff base hydrogels, four monoliths were obtained and their properties were compared. ADA15GEL20, ADA20GEL20, LYS15GEL20, and LYS20GEL20 were chosen for the test based on their storage modulus properties. To understand the different self-healing properties, the monoliths were cut into three pieces and immediately kept in contact with each other for 7 days and stored at 6 °C. Although all rebuilt monoliths were stable upon hanging, bending, and bridge formation, both hydrogels that contained more gelatin than the alginate derivate (ADA15GEL20 and LYS15GEL20) showed better healing at the cut interfaces, as seen in Figure 6 and Appendix A. Increasing the amount of ADA or f−ADA was reported to increase the crosslinking degree of their hydrogels (Table 2) [31].

Three pieces in all four monolith samples were found to be joined together after 7 days of resting (Appendix A). The resting time was provided for the hydrogels to heal after reforming the hydrogen bonding and the electrostatic repulsion. Three of the four samples were able to form a stable self-supporting bridge, allowing hanging and bending (Figure 7 and Appendix A). The ADA20GEL20 hydrogel was able to join back together like the rest, but not during bending. From the monolith test, we found that the ratio of gelatin being used in the hydrogel formula is the most determining factor for their ability to heal toward mechanical stresses (hanging, bending, and bridge formation). Likewise, the additional l-lysine residue on f-ADA was also contributed to the healing properties of the hydrogels. In summary, it is noteworthy to mention that both the alginate residue and gelatin affected the crosslinking degree and the mechanical properties of the dynamic covalent hydrogels.

## 4. Conclusions

The l-lysine alginate dialdehyde was successfully synthesized through partial periodate oxidation and reductive amination. The obtained values for the degrees of oxidation and substitution were 31 mol% and 20 mol%, respectively. The structural change in the polymer was observed via FTIR spectroscopy, confirming the formation of aldehyde groups as a result of reductive amination and Schiff base covalent linkage after the crosslinking. Such dynamic covalent crosslinking yields hydrogels with the ability to self-repair upon mechanical damage. The results indicate that the correlation between the concentration and stoichiometric ratio of polymers and gelatin is significant to attain preferable shear-thinning and self-healing properties in hydrogels. The viscosity of the hydrogel samples was affected mainly by the gelatin concentration. At the same time, the storage and loss moduli were also influenced by the molecular weight of the alginate derivatives and l-lysine residue. The results of this study could be relevant for further research on the application of functionalized alginate dialdehyde–gelatin hydrogels in drug delivery systems.

## Figures and Tables

**Figure 1 polymers-15-01010-f001:**
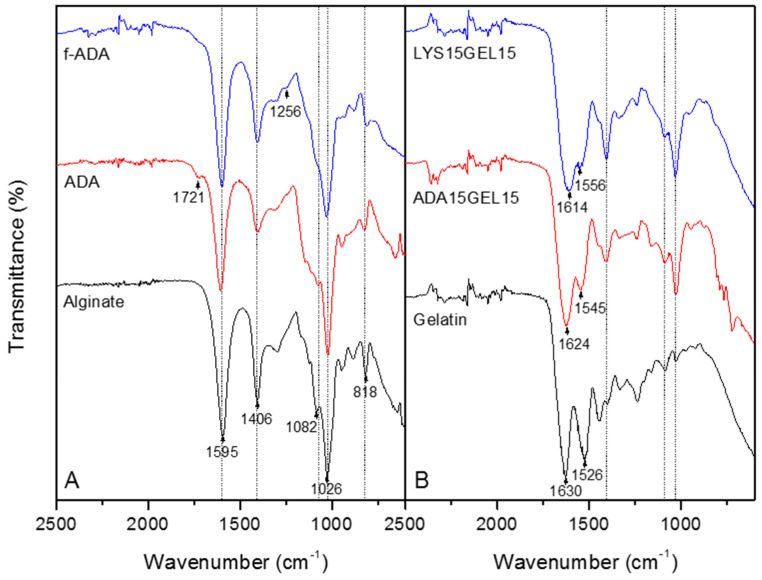
ATR−FTIR spectra of sodium alginate, ADA, and l-lysine−functionalized ADA (**A**) and ADA15GEL15, LYS15GEL15, and gelatin (**B**). The dash represents the alginate’s characteristic absorption peak.

**Figure 2 polymers-15-01010-f002:**
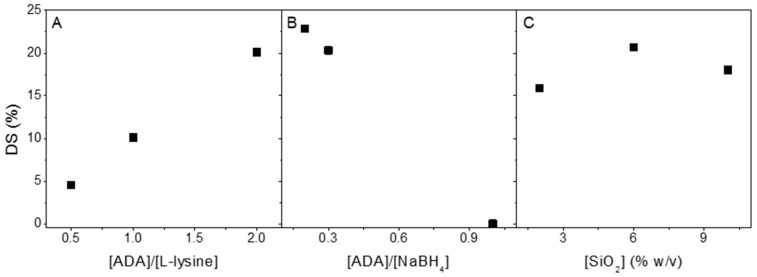
Degree of substitution (mmol %) as a function of the concentration of l-lysine (mmol) using 3 mmol of NaBH_4_ and 6% (*w*/*v*) SiO_2_ (**A**), NaBH_4_ (mmol) using 1 mmol of ADA and 6% (*w*/*v*) SiO_2_ (**B**), and 2,6, and 10% (*w*/*v*) SiO_2_ using 1 mmol of ADA and 3 mmol of NaBH_4_ (**C**). All reactions were conducted at pH 5.8 at room temperature in 10% methanol solutions.

**Figure 3 polymers-15-01010-f003:**
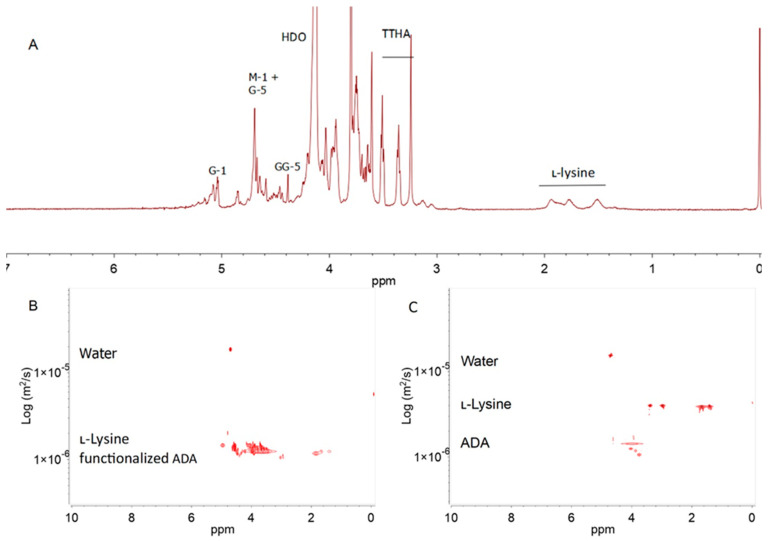
The ^1^H NMR spectrum (500 MHz, 90 °C) of l-lysine−functionalized ADA in D_2_O with the identification of the main signals. The spectral annotation of the alginate residue is given according to Grasdalen et al. [5] (**A**). The DOSY spectra of a mixture of non−covalently bound ADA and l-lysine (**B**) and l-lysine-functionalized ADA (**C**) in D_2_O recorded at 25 °C.

**Figure 4 polymers-15-01010-f004:**
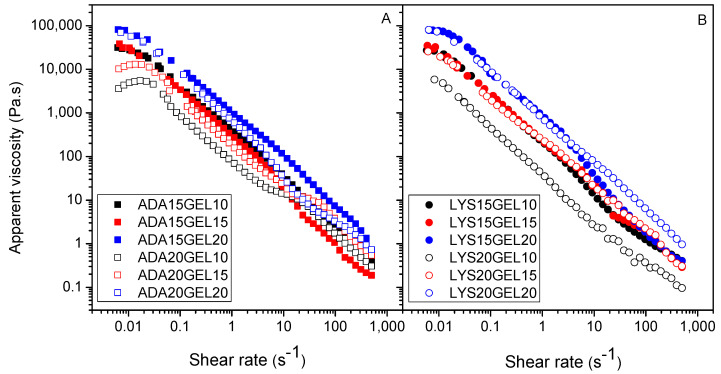
Apparent viscosity of ADA hydrogels (**A**), and l-lysine functionalized ADA hydrogels (**B**) with different amount of ADA/f−ADA and gelatin.

**Figure 5 polymers-15-01010-f005:**
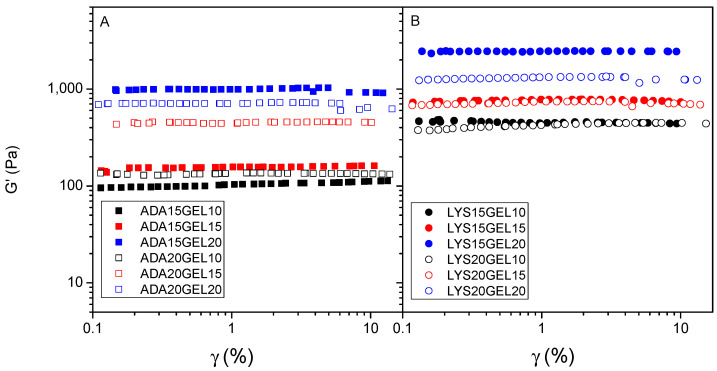
Amplitude strain sweep for hydrogels of alginate derivate samples crosslinked with 10 (black), 15 (red), and 20 (blue) (% *w*/*v*) of gelatin: 15% (*w*/*v*) of ADA (filled square) and 20% (*w*/*v*) of ADA (square) (**A**); 15% (*w*/*v*) of l-lysine−functionalized ADA (filled circle) and 20% (*w*/*v*) of l-lysine−functionalized ADA (circle) (**B**).

**Figure 6 polymers-15-01010-f006:**
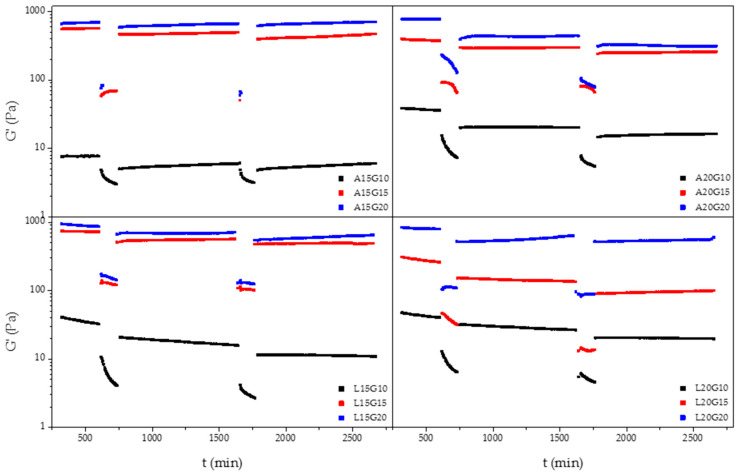
Representative 3−step loop rheological thixotropic test for hydrogels with difference amounts of ADA, f−ADA, and gelatin.

**Figure 7 polymers-15-01010-f007:**
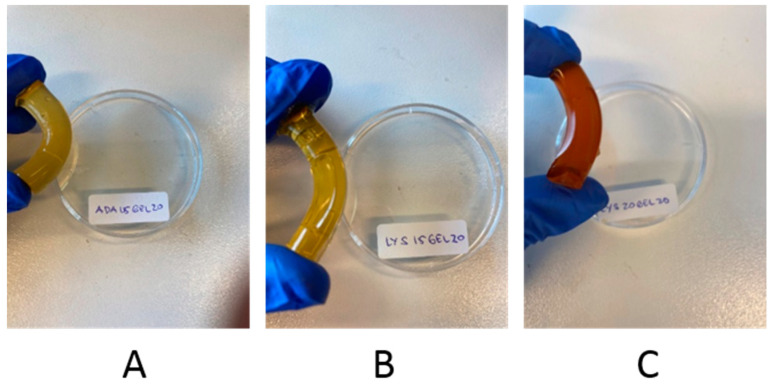
Photographs showing the resistance of self-healed hydrogels to bending: ADA15GEL20 (**A**); LYS15GEL20 (**B**); LYS20GEL20 (**C**).

**Table 1 polymers-15-01010-t001:** An overview on the research into the functionalization of alginate dialdehyde.

Functionalization Method	Modifier Molecule	Biopolymer Product	Ref.
Reductive amination (NaBH_3_CN)	l-Cysteine	S-protected thiolated alginate	[19]
Reductive amination (Pic-BH_3_)	Peptide sequence (GRGDYP, GRGDSP, and KHIFSDDSSE	Peptide coupled alginate	[20]
Carbodiimide chemistry	Norbornene/tetrazine and RGD peptides	Hydrolytically degradable click-crosslinked alginate hydrogels	[21]
Reductive amination (NaBH_3_CN)	Chitosan	pH-responsive multilayer film	[22]
Reductive amination (Pic-BH_3_)	4-(2-aminoethyl)benzoic acid	pH-tunable hydrogels	[23]
Reductive amination (NaBH_4_)	Cysteamine	Disulphide crosslinked alginate nanospheres	[24]
Click reaction	Cyclodextrin	Prolonged released alginate hydrogels	[17]
Reductive amination (NaBH_3_CN)	Alkyl amine	Self-assembled alginate microcapsule	[25]
Reductive amination (NaBH_4_)	Cysteine	Alginate–cysteine conjugate molecule	[26]
Reductive amination (NaBH_3_CN)	Octylamine	Alginate electrospun composite nanofiber	[27]
Thiol-aldehyde addition reaction	Glutathione modified sodium alginate	Alginate injectible hydrogels	[28]
Ugi-multicoponent reaction	Octylamine	Alginate self-assembled nanoparticles	[29]

**Table 2 polymers-15-01010-t002:** The GPC results for alginate samples, the degrees of oxidation of ADA, and the degrees of substitution of f-ADA.

Sample	M_n_ ^a^ (g/mol)	M_w_ ^a^ (g/mol)	PDI ^b^	Degree of Oxidation ^c^	Degree of Substitution ^d^	N ^e^ (%)
Alginate	110,100	358,900	3.25	-	-	-
ADA	35,280	64,800	1.81	0.31 ± 0.02	-	-
f-ADA	10,680	22,410	2.09	-	20.1 ± 3.2	2.47

^a^ Determined by GPC; ^b^ polydispersity index defined as Mw/Mn; ^c^ defined in Equation (1); ^d^ defined in Equation (2); ^e^ N content in the samples (wt%) was obtained using elemental analyses.

**Table 3 polymers-15-01010-t003:** Labels are used for the different samples as a function of their composition and crosslinking degree.

Concentration (*w*/*v*)%		Labels for Compositions		Degree of Crosslinking (%) ^a^	
ADA/f−ADA	Gelatin	ADA	f−ADA	ADA	f−ADA
15	10	ADA15GEL10	LYS15GEL10	29.1 ± 6	30.4 ± 4
15	15	ADA15GEL15	LYS15GEL15	40.6 ± 10	41.1 ± 7
15	20	ADA15GEL20	LYS15GEL20	49.8 ± 5	47.8 ± 3
20	10	ADA20GEL10	LYS20GEL10	37.9 ± 8	36.6 ± 1
20	15	ADA20GEL15	LYS20GEL15	43.5 ± 3	45.9 ± 4
20	20	ADA20GEL20	LYS20GEL20	55.4 ± 4	60.7 ± 4

^a^ Defined in Equation (3).

## Data Availability

Research data pertinent to this work are available upon request.

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
