# Peer review of "Rheological and Self-Healing Behavior of Hydrogels Synthesized from l-Lysine-Functionalized Alginate Dialdehyde"

_polymers, 2023, doi:10.3390/polym15041010_

Round 1

Reviewer 1 Report

The article by Putri et al deals with hydrogels of ÊŸ-lysine functionalized alginate dialdehyde that could have application in drug delivery systems. The topic is interesting and suitable for the journal readers. The authors have carried out a compelte characterization of the hydrogels. Some points should be addressed prior potential publication.

1)    SEM analysis of the hydrogels is recommended

2)    Some sentences should be rephrases: for instance their polysaccharide structure's characteristic absorption bands; suggests this group's involvement in the crosslinking reaction

3)    In figure 3, the whole FTIR spectra should be shown, thought a cut of the axis can be performed is necessary

4)    Figure 5 should be further explained

5)    Measurements of the storage modulus as a function of time should be carried out

6)    A table of comparison with previous studies incorporating alginate dialdehyde should be provided to support the improvements of the current study.

Reviewer 2 Report

The authors researched the rheological and self-healing behavior of hydrogels synthesized from ÊŸ-lysine functionalized alginate dialdehyde. The concluded results are valuable. However, some issues must be addressed before publication.

1. Make sure all abbreviations are written out in full the first time used. Please check the manuscript carefully.

2. It is recommended that any extra spaces in the full text be checked and removed.

3. There are some mistakes and misused words in the manuscript. Therefore, English must be polished by an expert.

4. Revising the references. There may be some references cited that do not help the reader to follow your work and results, nor to assess your approach or place your work or results within the state of the art.

Reviewer 3 Report

1. Isn't it already possible to form Schiff base reaction between alginate dialdehyde and gelatin to get imine bond? Therefore, the need for L-lysine accession has to be described. For example, the introduction of L-lysine brings about a change in the mechanics/biology of the function.

2. The Abstract section should be clear and concise: the important results and main conclusions drawn in this paper should be highlighted and presented in more precise language. For example, lines 69–70, there is no connection between the description of the rheology and the preceding and following texts.

3. Where are these hydrogels going to be used in real life? Advantages of the designed hydrogel can be improved by comparing and citing 10.1016/j.carbpol.2021.118046. The novelty of this work can be described at the end of the Introduction.

4. There are some formatting errors in the article. For example, spelling of references must be checked to meet the journal style (such as Reference 18). Please check carefully and use it properly.

5. Figure 5, how to prepare the gel samples for rheometry? The rheometric recording conditions should be detailed because the rheometric data are affected by many parameters. For example, before the dynamic frequency sweep assay, the linear viscoelastic range of the hydrogel should be tested. I want some comments to be made to address this point.

6. Background descriptions for hydrogel self-healing can be strengthened by citing 10.1016/j.cej.2022.135691 and what are the advantages of the current work compared to published articles?

7. The manuscript lacks any information on experimental replication. This is particularly worrisome. Please revise the manuscript detailing your experimental and technical replications.

Round 2

Reviewer 1 Report

My main concerns have been addressed properly

Reviewer 2 Report

It can be accepted.